# Factors Determining Soil Water Repellency in Two Coniferous Plantations on a Hillslope

**Moein Farahnak** [1], **Keiji Mitsuyasu** [1], **Kyoichi Otsuki** [2], **Kuniyoshi Shimizu** [3] and **Atsushi Kume** [3,*]

1   Graduate School of Bioresources and Bioenvironmental Science, Faculty of Agriculture, Kyushu University, Fukuoka 8190395, Japan
2   Kyushu University Forest, Kyushu University, Fukuoka 8112415, Japan
3   Faculty of Agriculture, Kyushu University, Fukuoka 8190395, Japan
*   Correspondence: akume@agr.kyushu-u.ac.jp; Tel.: +81-92-802-4674

**Abstract:** Soil water repellency (SWR) is a cause of low water infiltration, overland flow and soil erosion in mountainous coniferous plantations in Japan. The factors determining SWR intensity were investigated in two coniferous plantations of *Chamaecyparis obtusa* (Siebold et Zucc.) Endl. and *Cryptomeria japonica* (L.f.) D. Don, using intact tree plots and cut tree plots on the same hillslope. The SWR of *Ch. obtusa* plots was stronger than that of *Cr. japonica* plots. SWR intensity decreased after tree cutting. There were no significant differences in SWR upslope and downslope of individual trees/stumps for both tree species, though areas downslope of individual *Ch. obtusa* trees had higher SWR intensity than those upslope. SWR intensity and soil aggregate stability were positively correlated in the *Ch. obtusa* intact tree plot ($r = 0.88$, $p < 0.01$), whereas in the cut tree plot, this correlation was weak with no significance ($r = 0.29$, $p = 0.41$). Soil aggregate size had a non-significant influence on SWR intensity. These findings suggest that SWR intensity was not related to the soil aggregate size, but SWR intensity seemed have a role in soil aggregation in the *Ch. obtusa* intact tree plot. Destruction of soil aggregates could occur after tree cutting because of physical disturbances or increased input of different types of organic matter from other vegetation into soil. The presence of *Ch. obtusa* introduces a source of SWR, although uncertainty remains about how water repellency is distributed around soil aggregates. The distribution pattern of soil water content and soil hydraulic conductivity around *Cr. japonica* was related to other factors such as the litter layer and non-water-repellant soil.

**Keywords:** *Chamaecyparis obtuse*; *Cryptomeria japonica*; mean weight diameter; soil moisture; soil aggregate size; soil moisture; soil water repellency; hill slope

## 1. Introduction

Soil water repellency (SWR) is a phenomenon that delays water penetration into the soil [1]. This phenomenon decreases water infiltration and soil hydraulic conductivity [2–5]. Decreases in soil hydraulic properties subsequently enhance overland flow and soil erosion [3–5]. In general, specific biotic and abiotic factors are responsible for SWR. These factors consist of the type of tree species [6–8], the fungal communities near tree roots [9], soil organic matter [10], soil aggregate size [11] and soil water content [12]. In particular, low infiltration, overland flow and soil erosion have been reported in mountainous plantations where the soil surface was coated with water-repellent substances [3–5,13–15]. Although the possible impact of these properties on SWR intensity has been well discussed in previous studies [9–20], it remains unclear how these soil properties are distributed around trees on a slope. A standing tree on a slope provides different conditions in the areas upslope and downslope of its

trunk [2]. Both the organic and inorganic layers of soil in these two areas are influenced by the standing tree [2,21]. Liang et al. [21,22] showed that there was a larger amount of stemflow and soil water content on the downslope side of a broadleaved tree (*Stewartia monadelpha*) than on the upslope side, which was related to unequal canopy coverage and tree tilting downslope. They also investigated the period of dryness around *S. monadelpha* and identified stemflow and the root-induced pathway (double-funneling) as the two main reasons for fast dryness downslope of a tree [22]. Farahnak et al. [2] found different soil hydraulic conductivities between areas upslope and downslope of two coniferous tree species on the same slope. They demonstrated that different biological aspects influenced soil hydraulic conductivity around individual *Cryptomeria japonica* and *Chamaecyparis obtusa* trees on the slope [2]. Litter biomass accumulation (upslope of *Cr. japonica*) and SWR (downslope of *Ch. obtusa*) were responsible for increasing and decreasing soil hydraulic conductivity, respectively [2]. It was observed that changes in water supply (i.e., removal of aboveground biomass) in cut tree plots did not change soil hydraulic conductivity substantially around individual stumps of *Cr. japonica* and *Ch. obtusa*, despite the reduction in SWR intensity around individual stumps of *Ch. obtusa* [2].

In general, soil water content is negatively correlated with the severity of SWR [11]; in other words, SWR intensity depends on the patterns of soil moisture distribution and these patterns might be different around individual trees or stumps in slope areas [2,21]. Therefore, we assumed that SWR might have different distributions and intensities upslope and downslope of individual trees and stumps with variations in soil properties in these areas.

Forest operations such as clear cutting and thinning in sloping areas lead to substantial changes in water supplies and soil properties [23–27]; however, no previous studies have shown how SWR intensity alters after tree cutting. The specific effects of vegetation on nearby soil might provide clearer information about SWR distribution and possible related overland flow and other soil hydraulic properties in tree-scale studies. However, with increases in study scale, the determinants of SWR distribution also increase, which may make it difficult to obtain a definite conclusion.

In this study, we investigated SWR intensity between two adjacent coniferous plantations on the same hillslope. We compared plots with intact trees and cut trees to understand the impact of tree cutting on SWR intensity. The main aim of the study was to elucidate the potential soil properties that affect SWR intensity by considering the type of tree species, the presence or absence of tree cutting (i.e., removal from aboveground), and position (i.e., upslope and downslope) around individual trees and stumps. We selected these areas as they are most strongly influenced by individual trees or stumps in slope areas. The areas upslope and downslope of standing trees are important areas that receive the greatest input of water from aboveground. Understanding the causes of prevention of water input (i.e., SWR) and the intensity of these causes may be crucial to estimating overland flow. In addition, changes in SWR intensity or factors that control SWR intensity around individual stumps might clarify the alteration in water penetration or overland flow in slope areas used for forestry. Hence, we hypothesized that: (1) SWR intensity is species-specific; (2) different soil properties between the upslope and downslope of intact tree and cut tree plots induce different SWR intensities; and (3) SWR intensity is influenced by the size of soil aggregates, which influence the hydraulic conductivity.

## 2. Materials and Methods

### 2.1. Study Site and Sampling Design

This study was conducted in two adjacent coniferous plantations of *Chamaecyparis obtusa* (Siebold et Zucc.) Endl. and *Cryptomeria japonica* (L.f.) D. Don in the northern part of Kyushu, Japan (33°38'N, 130°31'E). *Ch. obtusa* and *Cr. japonica* occupy 25% and 44%, respectively, of plantations in Japan [28]. *Ch. obtusa* and *Cr. japonica* were planted in 1961 (5.2 ha) and 1934 (0.5 ha), respectively. Parts of these plantations were logged from December 2015 to February 2016, at which time clear-cutting (*Ch. obtusa*) and thinning (*Cr. japonica*) were conducted (1.14 ha). The current study was conducted in 2017, one year after logging. The soil type is classified as brown forest soil originating from serpentine

bedrock [2]. The mean annual air temperature and precipitation (2007–2016) of the meteorological station located 15 km southwest of the study site were 17 °C and 1700 mm, respectively (Hakata Meteorological Station, 33°36′N, 130°25′E, ca. 3 m above sea level (a.s.l.)). Both plantations have a density of 1600 trees ha$^{-1}$. In total, four study plots were chosen: (1) *Ch. obtusa* intact tree plot; (2) *Cr. japonica* intact tree plot; (3) *Ch. obtusa* cut tree plot; and (4) *Cr. japonica* cut tree plot. The intact and cut tree plots were selected to possess the same site characteristics (slope 30°, southwest facing, and 130 m a.s.l; see [2]). In each plot, we selected five individual trees or stumps for collection of soil samples. Soil samples at 0–5 cm depth were taken at a distance of 0.5 m upslope and downslope from each individual tree/stump (Figure 1). Thus, there were five replicates for each individual position either upslope or downslope of individual trees or stumps. A total of 10 replicates (5 upslope and 5 downslope) were collected in each plot. We also collected soil samples from 5–10 and 10–30 cm depths to evaluate SWR at different depths. The average distance from other individual trees or stumps in the upslope and downslope areas of selected trees and stumps was ~4 m in *Ch. obtusa* plots and ~3 m in *Cr. japonica* plots. Two sets of soil samples were collected to measure SWR intensity and soil properties. Soil texture was determined by the hydrometer method. Intact soil samples were collected with a steel cylinder (100 cm$^3$) for determination of soil bulk density and soil porosity. Soil organic matter (SOM) was measured using the loss on ignition method. Soil gravimetric water content ($\theta_g$) and aggregate mean weight diameter (MWD) were also measured, as described below. The soil samples were transferred into the laboratory within 24 h and were air-dried at room temperature (20 °C) for about 1 week (Table 1; for further details on the study site, see [2]).

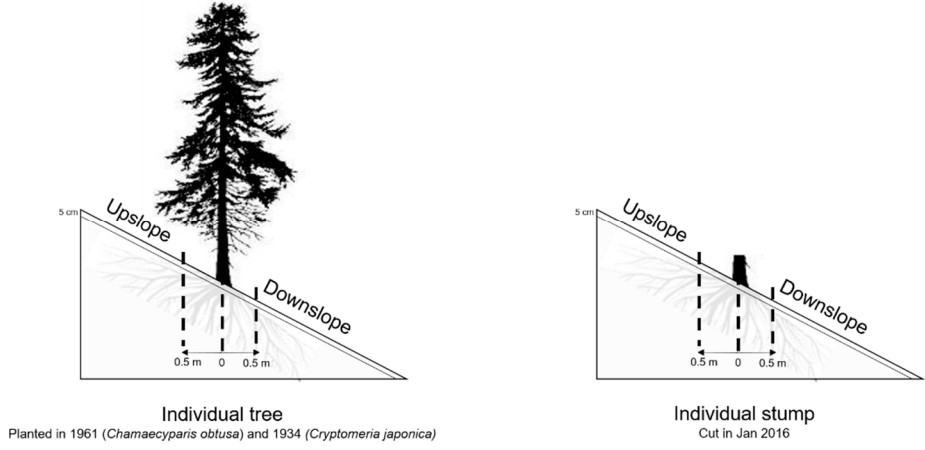

**Figure 1.** Sampling design around individual trees and stumps.

**Table 1.** Basic soil characteristics.

| Tree Species | Plot | Sand | Silt | Clay | Soil Texture | Soil Bulk Density | Soil Porosity |
|---|---|---|---|---|---|---|---|
| | | (%) | | | | (g cm$^{-3}$) | (%) |
| *Chamaecyparis obtusa* | Intact tree | 75.14 | 15.59 | 09.28 | Loam | 0.87 ± 0.02 | 65.59 ± 0.91 |
| | Cut tree | 73.08 | 15.14 | 11.78 | Loamy sand | 0.87 ± 0.04 | 65.60 ± 1.73 |
| *Cryptomeria japonica* | Intact tree | 63.13 | 25.92 | 10.94 | Loamy sand | 1.02 ± 0.03 | 59.88 ± 1.35 |
| | Cut tree | 70.03 | 20.24 | 09.72 | Silty loam | 0.97 ± 0.03 | 62.18 ± 1.24 |

Values are the mean ± standard error (n = 10, soil depth = 5 cm); data for soil texture, soil bulk density, and soil porosity were retrieved from [21].

## 2.2. Sample Preparation for SWR Tests

Kobayashi [29] indicated that experimental conditions such as initial soil water content and room temperature affect SWR intensity. Therefore, we tried to minimize variation in the moisture conditions by keeping all soil samples at the same temperature (20 °C, room temperature regulated with an air

conditioner). Four types of subsamples were prepared for SWR testing: (1) air-dried soil samples; (2) samples from different soil depths; (3) soil samples sieved to different soil aggregate sizes; and (4) broken soil aggregate samples. Air-dried samples were passed through a 2.00-mm sieve, after which they were used for the water repellency test. The SWR was also tested for 0–5, 5–10 and 10–30 cm soil depths to assess the SWR intensity through the soil profile. To evaluate the effects of soil aggregate size, the air-dried soil samples were sieved at five different sieve sizes (mesh size: 2.00, 1.00, 0.50, 0.25 and 0.10 mm) by dry sieving [30]. These samples were also tested separately for SWR intensity. The extra 2.00-mm sieved soil samples were crushed gently by a mortar and pestle to evaluate the impact of broken soil aggregates on SWR intensity.

### 2.3. Molarity of an Ethanol Droplet Test

In this study, the molarity of an ethanol droplet (MED) test was used for evaluating SWR intensity (%) [31]. The MED test measures the potential SWR intensity of the soil surface by dropping various ethanol dilutions on the soil surface and observing their infiltration performance. Pure ethanol has a lower surface tension (22.7 mN m$^{-1}$ at 20 °C) than water (72.8 mN m$^{-1}$ at 20 °C). Dilution of pure ethanol with water at different percentages provides different surface tension values. The surface tension of a droplet on the soil surface decreases with increasing ethanol percentage, and vice versa. In our experiment, the pure ethanol was diluted with distilled water into separate solutions containing different percentages of ethanol on a volume basis (0%, 1.0%, 3.0%, 5.0%, 8.5%, 13.0%, 18.0%, 24.0% and 36.0%, v v$^{-1}$) [31]. The SWR intensity was tested in soil samples by dropping these solutions onto surfaces of smoothed soil samples in Petri dishes using a micropipette (droplet size: 50.0 µL) from 5.0 mm height in increasing order of concentration, from 0% to 36%. When a droplet of specific solution infiltrated the soil within 3 s, the solution before that was selected as the potential SWR intensity (except 0%). For example, if the 8.5% droplet infiltrated within 3 s, the 5.0% solution was selected as the solution defining the SWR intensity of that soil sample (Figure 2). In this study, we present the ethanol % in the results to show the SWR intensity of soil samples (with increasing ethanol %, SWR intensity also increases).

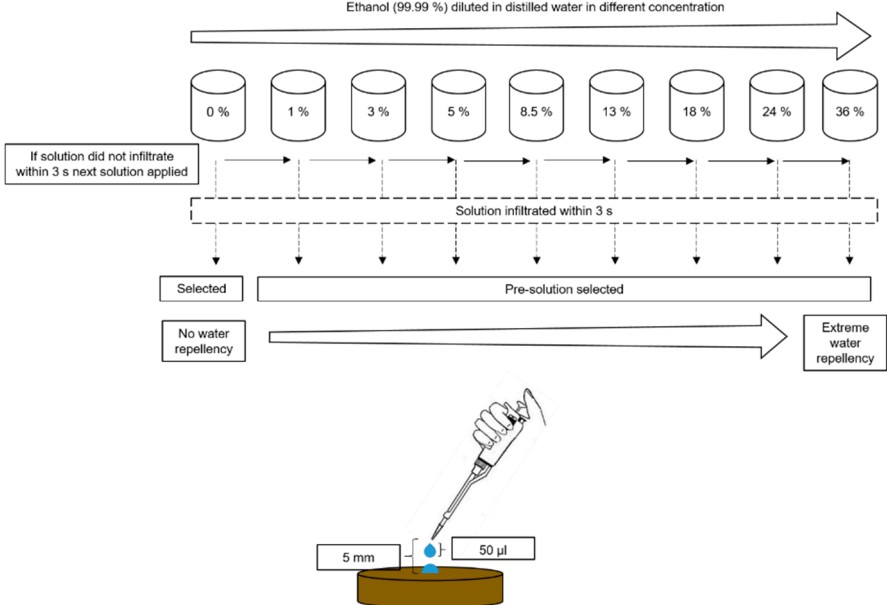

**Figure 2.** Schematic illustration of the molarity of an ethanol droplet (MED) test.

### 2.4. Soil Water and Moisture Content Measurement

Soil gravimetric water content ($\theta_g$, g g$^{-1}$) of air-dried samples was measured [32]. Approximately 10 g air-dried-soil samples were weighed before and after drying for 24 h at 110°C. The following equation was used:

$$\theta_g = \frac{(\text{IW} - \text{DW})}{\text{DW}} \tag{1}$$

where $\theta_g$ is the soil gravimetric water content of air-dried-soil, and IW and DW are the initial (air-dried) and oven-dried mass of soil samples (g), respectively.

In the field, two analog capacitive soil moisture sensors (S-SMC-M005; Onset Computer Corporation, Bourne, MA, USA) were installed (at 45° angles) at 5 cm depth in the soil at 0.5 m upslope and downslope from individual trees/stumps for which the soil was not sampled (eight sensors in total, two in each plot). Prior to installation, the soil sensors were calibrated by testing them in dried sandy soil, saturated sandy soil, and tap water to check their accuracy. Two sensors in each plot were connected to a logger (HOBO RX3000; Onset Computer Corporation, Bourne, MA, USA) to record hourly soil moisture content (m$^3$ m$^{-3}$) from July 13 to July 27, 2018, during which period one rain event was recorded (Figure 3). Soil moisture content data were collected at midnight (00:00) because of data fluctuations during the daytime in response to changes in soil temperature, especially in the cut tree plots.

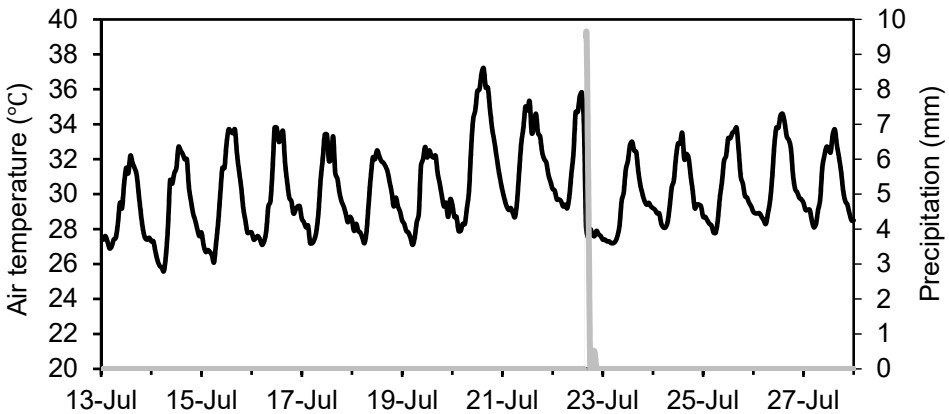

**Figure 3.** Precipitation (mm) and air temperature (°C) records at Hakata Meteorological Station between 13 July 2018 and 27 July 2018. Black and gray lines represent air temperature and precipitation, respectively.

### 2.5. Soil Aggregate Stability

To measure the degree of soil structure development, aggregate mean weight diameter (MWD, mm) was calculated as a statistical index of soil aggregative stability by the wet sieving method [33]. Air-dried samples from 0–5 cm soil depth were sieved (with 8.00 mm and then 2.00 mm sieves) to limit the aggregate size. Soil samples (<2.00 mm) were sprayed with distilled water three times at intervals of 30 min and then immersed in distilled water for 24 h to prevent aggregate slaking. These samples were put into a sieve set (mesh diameters: 2.00, 1.00, 0.50, 0.25, and 0.10 mm) and divided into five fractions by vertical shaking that was repeated 150 times in water. The soil suspension in each sieve was filtered (cotton cellulose paper filter with 5.00 μm pore size) and the remaining soil was weighed after 24 h oven-drying. After measuring the dried mass of aggregates, aggregates of each size were broken by pouring hot water on the corresponding sieves. The remaining coarse aggregates were oven-dried for 24 h and weighed by subtracting the initial mass of aggregates. A cumulative frequency curve is obtained when the cumulative percentage weights are plotted against the upper

limits of separation (i.e., mesh size). To summarize the distribution in one single value, we calculated the MWD as follows:

$$F(x) = \int_0^x f(x) \, dx \tag{2}$$

where $x$ is the diameter of the aggregates, F $(x)$ is the cumulative frequency function, and f $(x)$ is the frequency density function. The MWD is defined as $\int_0^{8.00} f(x) \, dx$. By partial integration: $\int_0^{8.00} x f(x) \, dx = \{x F(x)\}_0^{8.00} - \int_0^{8.00} F(x) \, dx$. $\int_0^{8.00} F(x) \, dx$ is the area under the curve F $(x)$, which can be measured and subtracted from 8 to obtain the mean diameter [33].

*2.6. Statistical Analysis*

The mean ± standard error (SE) of all data were presented. Generalized linear modeling (Poisson, log) was applied to fit the best model to find the effect of tree species, cutting and position on SWR intensity in all data (number of samples = 40). We also used one-way analysis of variance (ANOVA) and the Tukey's HSD test to detect the differences between SWR intensity between upslope and downslope areas with five replicates in each plot. We tested for significant differences between SWR intensity at different soil depths, SWR intensity between different soil aggregate sizes, and SWR intensity before and after breaking soil aggregates with five and ten replicates by ANOVA and the Tukey's_HSD test. Pearson's correlation analysis ($r$) was performed between SWR and SOM, $\theta_g$ and MWD for all five and ten replicates. All statistical analyses were conducted in RStudio software version 1.0.153 (Rstudio, Inc., Boston, MA, USA).

## 3. Results

The results for the effect of tree species, cutting and position on SWR intensity and the differences in SWR intensity upslope and downslope of individual *Ch. obtusa* and *Cr. japonica* trees and stumps are plotted in Figure 4. *Ch. obtusa* had higher SWR intensity than *Cr. japonica,* and forest operation (cutting) lowered SWR intensity (Figure 4(a-1,2)). The position on the slope (upslope and downslope areas) did not show consistent effects on the SWR intensity (Figure 4(a-3)); however, the samples downslope of individual *Ch. obtusa* trees had higher SWR intensity than those upslope (Figure 4(b-1)).

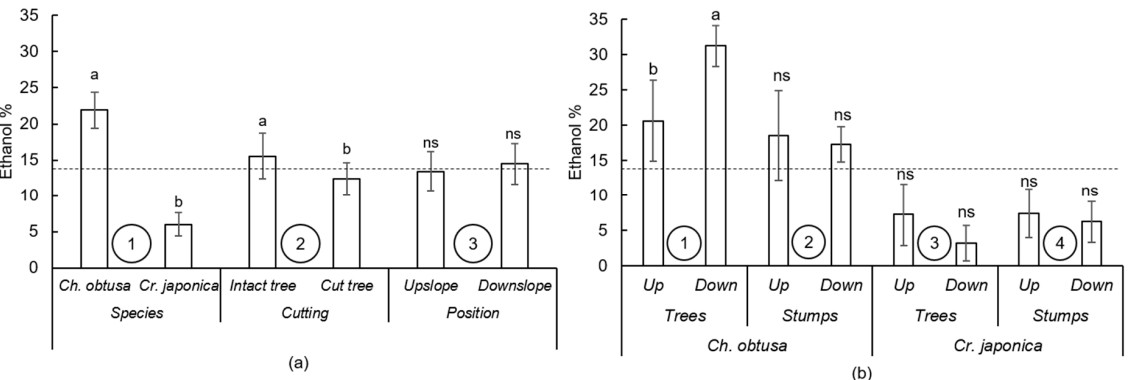

**Figure 4.** The effect of tree species (*Ch. obtusa* and *Cr. japonica*), cutting (intact tree and cut tree plots), and position (upslope and downslope) on soil water repellency (SWR) intensity (generalized linear model) (**a**). Comparison of SWR intensity between upslope and downslope of individual *Ch. obtusa* and *Cr. japonica* trees and stumps (**b**). Small letters and ns denote significant and non-significant differences in each treatment, respectively. Soil samples are from 5 cm depth. Error bars indicate the standard error (SE).

The MED test at different soil depths showed a decreasing trend in SWR intensity with increasing soil depth (Figure 5). The SWR intensity was significantly higher at 5 cm soil depth in *Ch. obtusa* plots than that at 10 and 30 cm soil depths (Figure 5a,c). In *Cr. japonica* plots, significant differences were

only detected in the cut tree plot, with higher SWR intensity at 5 cm soil depth than that at deeper soil depths, although the intensity was not as strong as for *Ch. obtusa* (Figure 5d). Physically broken soil aggregates showed a significant decrease in SWR intensity in the *Ch. obtusa* intact tree plot, whereas SWR intensity did not change significantly in the *Cr. japonica* plot (Figure 6).

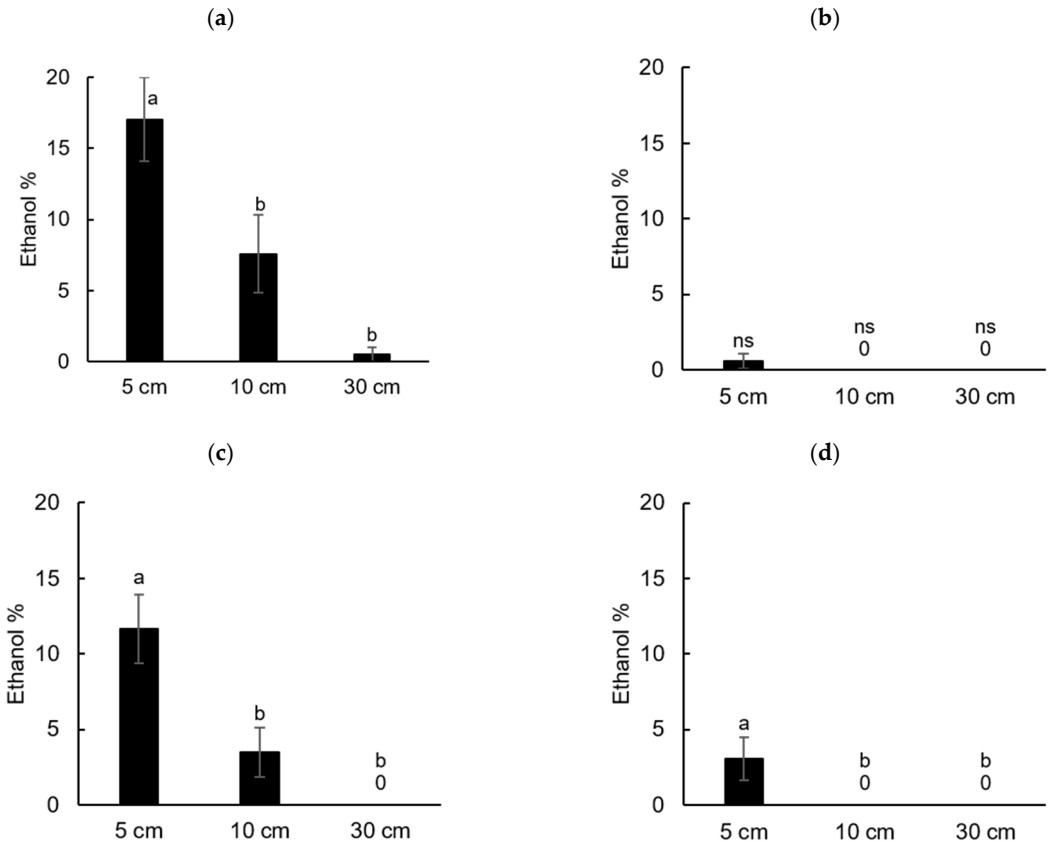

**Figure 5.** Soil water repellency (SWR) intensity at different soil depths in *Ch. obtusa* intact tree (**a**), *Cr. japonica* intact tree (**b**), *Ch. obtusa* cut tree (**c**) and *Cr. japonica* cut tree (**d**) plots. Small letters and ns denote significant and non-significant differences, respectively, between different soil depths.

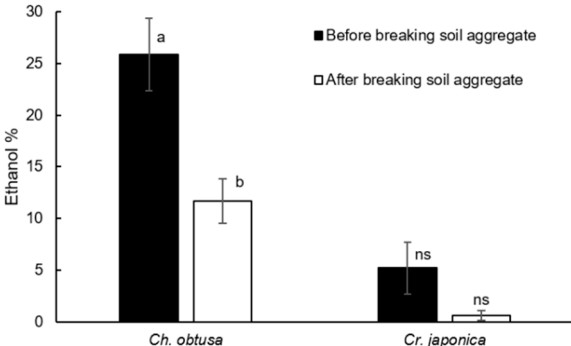

**Figure 6.** Soil water repellency (SWR) intensity of broken soil aggregates at 5 cm soil depth of *Ch. obtusa* and *Cr. japonica* intact tree plots. Small letters and ns denote significant and non-significant differences, respectively.

The soil properties in the study plots are summarized in Table 2. Soil properties did not display significant differences between soil upslope and downslope of individual trees and stumps in both tree species, though SOM was higher upslope than downslope of individual *Ch. obtusa* stumps (Table 2). The SOM and SWR intensity did not show any significant correlations (Table 3). The $\theta_g$ and SWR

intensity also did not show a significant correlation, and monitoring of soil moisture for two weeks with one rain event showed dissimilar patterns around individual *Ch. obtusa* and *Cr. japonica* trees and stumps (Figure 7). Soil moisture among individual *Ch. obtusa* trees showed different patterns, with higher soil moisture content upslope and lower moisture downslope (Figure 7a). The soil moisture upslope of individual *Ch. obtusa* trees increased after the rain event, whereas the downslope area did not show any changes (Figure 7a). The soil moisture exhibited a similar pattern and values upslope and downslope of individual *Ch. obtusa* stumps, and soil moisture increased in both areas after the rain event (Figure 7c). MWD showed a significant positive correlation with SWR intensity in the *Ch. obtusa* intact tree plot (Table 3). MWD had a positive correlation with SWR intensity on the upslope of individual *Ch. obtusa* trees. The correlation between MWD and SWR intensity was weak in the *Ch. obtusa* cut tree plot (Table 3). MWD correlation with SWR was also weak upslope of individual *Ch. obtusa* stumps (Table 3), which had the highest SOM content (Table 2).

**Table 2.** Soil organic matter (SOM), air-dried-soil gravimetric water content ($\theta_g$), and aggregate mean weighted diameter (MWD) from 5-cm soil depth of *Chamaecyparis obtusa* and *Cryptomeria japonica* intact tree and cut tree plots.

| Tree Species | Type | Position | SOM (%) | $\theta_g$ (g g$^{-1}$) | MWD (mm) |
|---|---|---|---|---|---|
| *Ch. obtusa* | Individual trees | U * | 13.62 ± 1.12 | 0.06 ± 0.00 | 5.39 ± 0.06 |
| | | D ** | 14.18 ± 1.59 | 0.06 ± 0.00 | 5.47 ± 0.03 |
| | | m *** | 13.90 ± 0.92 | 0.06 ± 0.00 | 5.43 ± 0.03 |
| | Individual stumps | U | 19.03 ± 3.97 | 0.05 ± 0.00 | 5.28 ± 0.09 |
| | | D | 11.81 ± 1.16 | 0.05 ± 0.00 | 5.39 ± 0.05 |
| | | m | 15.42 ± 2.29 | 0.05 ± 0.00 | 5.33 ± 0.05 |
| *Cr. japonica* | Individual trees | U | 14.96 ± 1.91 | 0.07 ± 0.01 | 4.93 ± 0.11 |
| | | D | 12.23 ± 1.03 | 0.04 ± 0.00 | 5.37 ± 0.02 |
| | | m | 13.60 ± 1.12 | 0.06 ± 0.01 | 5.15 ± 0.09 |
| | Individual stumps | U | 11.29 ± 1.25 | 0.08 ± 0.02 | 5.00 ± 0.03 |
| | | D | 10.06 ± 0.58 | 0.06 ± 0.00 | 5.04 ± 0.09 |
| | | m | 10.67 ± 0.68 | 0.07 ± 0.01 | 5.02 ± 0.05 |

\* U = upslope (5 samples), ** D = downslope (5 samples), *** m = mean of upslope and downslope areas (10 samples). SOM data were retrieved from [21].

**Table 3.** Pearson correlation coefficient (*r*) between soil water repellency (SWR) and soil properties (SOM, $\theta_g$ and MWD) upslope and downslope of individual *Chamaecyparis obtusa* and *Cryptomeria japonica* trees and stumps.

| Tree Species | Type | Position | SOM | $\theta_g$ | MWD |
|---|---|---|---|---|---|
| *Ch. obtusa* | Individual trees | U | −0.26 | 0.85 | 0.93 * |
| | | D | −0.51 | 0.14 | 0.59 |
| | | m | −0.23 | 0.52 | 0.88 * |
| | Individual stumps | U | 0.68 | 0.26 | 0.40 |
| | | D | 0.30 | 0.53 | 0.01 |
| | | m | 0.58 | 0.30 | 0.29 |
| *Cr. japonica* | Individual trees | U | −0.14 | 0.52 | 0.94 * |
| | | D | −0.63 | −0.23 | 0.05 |
| | | m | −0.11 | 0.47 | 0.25 |
| | Individual stumps | U | −0.65 | 0.42 | −0.51 |
| | | D | 0.25 | 0.38 | 0.23 |
| | | M | −0.33 | 0.36 | −0.00 |

SOM: soil organic matter, $\theta_g$: air-dried-soil gravimetric water content, MWD: aggregate mean weighted diameter, U = upslope (5 samples), D = downslope (5 samples), m = mean of upslope and downslope areas (10 samples) * significant correlation at $p < 0.05$. SOM data were retrieved from [21].

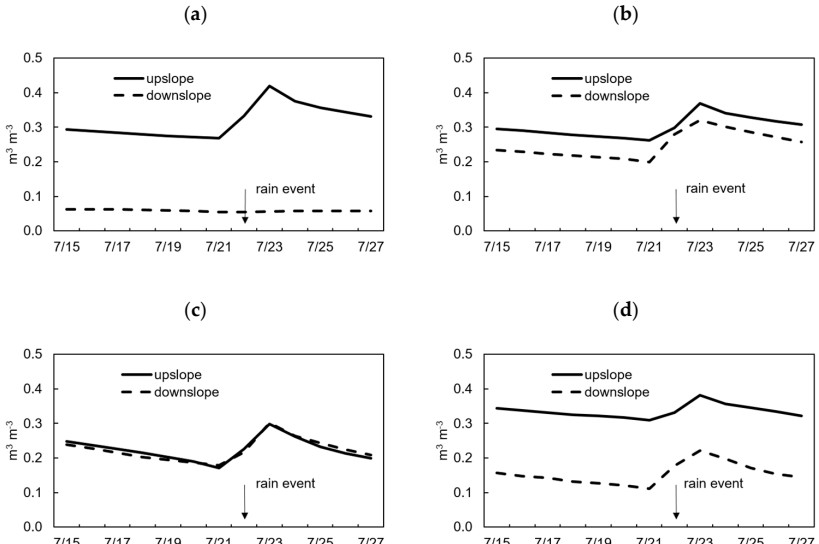

**Figure 7.** Soil moisture content (m³ m⁻³) at 5 cm soil depth upslope and downslope of individual *Ch. obtusa* trees (**a**), individual *Cr. japonica* trees (**b**), individual *Ch. obtusa* stumps (**c**) and individual *Cr. japonica* stumps (**d**). Down arrows indicate the rain event on 22 July.

The soil aggregate size measurements showed that the different soil particle sizes had non-significant effects on SWR intensity regardless of tree species and whether the trees in the plots were intact or had been cut (Figure 8). Soil aggregates showed higher SWR intensity in the *Ch. obtusa* plots than in the *Cr. japonica* plots (Figure 8a,b). The SWR intensity among all different soil particle sizes was also higher in the *Ch. obtusa* intact tree plot than in the *Ch. obtusa* cut tree plot (Figure 8). Soil aggregates downslope of individual *Ch. obtusa* trees had higher SWR intensity than those from upslope areas (Figure 8a,b). The SWR intensity among different sizes of soil aggregates was equal between upslope and downslope of individual *Ch. obtusa* stumps (Figure 8c,d).

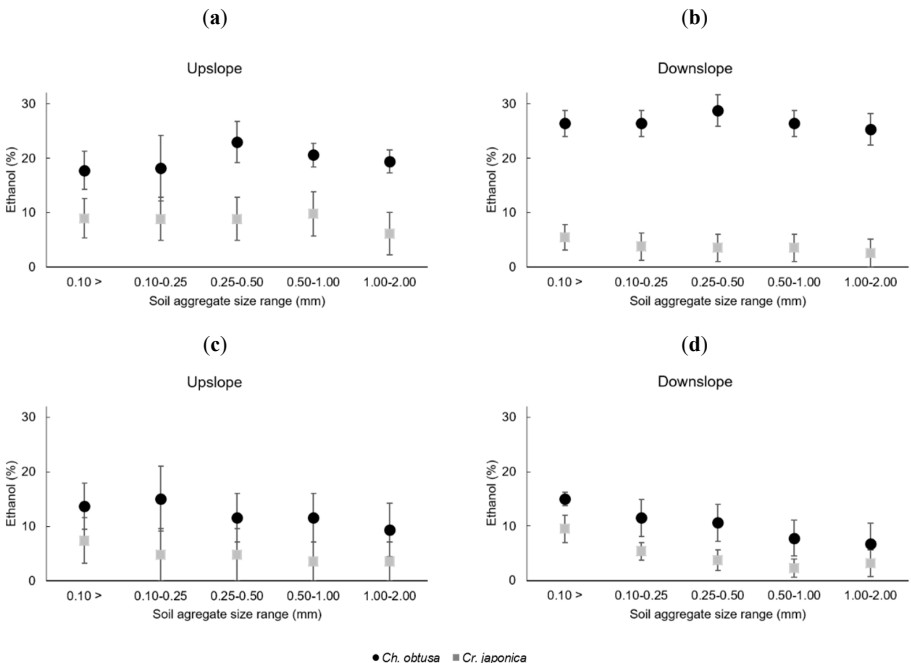

**Figure 8.** Soil water repellency (SWR) intensity at 5 cm soil depth of different soil particle sizes upslope (**a**) and downslope (**b**) of individual *Ch. obtusa* and *Cr. japonica* trees, and upslope (**c**) and downslope (**d**) of individual stumps.

## 4. Discussion

We attempted to examine the factors (such as type of tree species, forest operations, slope, and soil properties) that maintain the SWR of forest soil in two common types of Japanese plantations. The results clarify that SWR is maintained by the presence of specific tree species (i.e., *Ch. obtusa*) and is associated with the formation of soil structure. Previous researchers suggested that SWR is generated as a specific byproduct of tree species (mainly coniferous) in organic matter forms in forest areas [6,16,19,34]. Our study showed that *Ch. obtusa* widely supplies water-repellent materials to the forest-floor soil surface (Figure 4a). In contrast, low- or non-water-repellency of soil aggregates around individual *Cr. japonica* trees caused low SWR intensity and resulted in a homogenous high-water content in the soil [2]. However, as various soil characteristics (SOM, $\theta_g$, MWD and soil aggregate sizes) exhibited no significant spatial distribution, the mechanism by which the water-repellent material is maintained on the cypress forest floor is still unknown.

After tree cutting, the SWR intensity decreased, probably as a result of decomposition of the water-repellent material and runoff of water-repellent soil particles approximately one year after tree cutting. Soil erosion in cut tree plots could be one of the important factors for the decrease in SWR intensity one year after tree cutting (Figure 4). The high SWR intensity was mainly concentrated at 5 cm soil depth (Figure 5) and logging activities and overland flow easily disturbed the soil surface or removed material. The reduction of SWR intensity after breaking of soil aggregates is consistent with the possible effect of physical disturbances on soil aggregates covered with water-repelling substances (Figure 6).

Strong SWR intensity is one of the important factors for reduction of soil hydraulic conductivity and increases in overland flow at the small and large scales [3–5,35]. Our results also showed higher SWR intensity downslope of individual *Ch. obtusa* trees, which could have caused a reduction in soil moisture in this area during the monitoring period (Figure 7a). Soil moisture during two hot weeks in summer (Figure 3: mean air temperature: 30.2 °C) enhanced SWR intensity, especially downslope of individual *Ch. obtusa* trees (Figure 7a). Therefore, even after a rain event, because of the dry period and enhanced SWR intensity, soil moisture did not change downslope of individual *Ch. obtusa* trees (Figure 7a). Another factor that could have contributed to the spatial pattern of soil moisture is that downslope of individual *Ch. obtusa* trees, rainwater could not enter the soil because of the SWR, thus increasing the chance of overland flow. In contrast, in areas upslope of trees, water may remain by the tree trunk and slowly enter the soil despite the SWR (Figure 7a). The similar pattern of soil moisture upslope and downslope of individual *Ch. obtusa* stumps could be because the cut tree plot was directly exposed to solar radiation and thus had a high evaporation rate (Figure 7a). However, the distribution pattern of soil water content and soil hydraulic conductivity around individual *Cr. japonica* trees or stumps is likely related to other factors such as the litter layer and non-water repellent soil [2], as the SWR intensity did not show any significant spatial differences in these plots (Figure 4(b-3,4)).

The soil dryness and wetness pattern has been recognized as a factor that affects SWR intensity [12,36,37]. Increasing soil wetness reduces SWR intensity, whereas a long dry period enhances SWR intensity [12,36,37]. This pattern in the field can be explained by consistent input of rainwater, which results in a saturated soil matrix and weakened SWR prevention effects [12,36,37]. However, this is the case for moist soil; in air-dried samples, no correlation was detected between $\theta_g$ and SWR intensity (Table 2; Table 3). Soil moisture changes during the field monitoring showed different patterns upslope and downslope of individual trees and stumps (Figure 7). However, no significant relationship between these patterns of change and soil characteristics was found, suggesting that water infiltration on the forest soil surface depends on local soil characteristics rather than average soil characteristics. Liang et al. [22] and Kobayashi et al. [17] showed that root development plays an important role in soil water infiltration, so it was hypothesized that the water movement in the forest floor soil surface was controlled by the combination of roots and SWR intensity.

Some previous studies reported that SWR has the ability to enhance the stability of soil aggregates [9,34,38,39]. These studies explained how hydrophobic substances may become attached

to soil aggregates, acting as cement and causing increased soil aggregate stability and reduced swelling [9,38,39]. In this study, the positive correlation between SWR intensity and MWD explained the effect of SWR in increasing soil aggregate stability in the *Ch. obtusa* intact tree plot, especially upslope of individual *Ch. obtusa* trees (Table 3). The diminishing SWR and higher SOM in the *Ch. obtusa* plot also indicated the likely reduction in SWR intensity and destruction of soil aggregates by soil erosion and the possible production of different types of organic matter (higher SOM upslope of individual *Ch. obtusa* stumps) from annual plants in the *Ch. obtusa* cut tree plot (Table 2; Table 3).

Doerr et al. [9] reported that coarse and fine soil aggregates can influence SWR intensity. They explained that coarse soil aggregates (sandy) with low surface areas have more spaces for settling of hydrophobic substances and therefore have an increased SWR intensity [9]. Fine soil aggregates, however, are also reported to have high SWR intensity, which is possibly related to the higher surface areas of fine soils available to be covered with and to retain hydrophobic substances [9]. Our results showed a non-significant correlation of different soil aggregate sizes with SWR intensity, whereas soil aggregates from around *Ch. obtusa* had a higher SWR intensity than those from around *Cr. japonica* (Figure 8). This can be explained by the probable greater supply of hydrophobic substances from *Ch. obtusa*, which settle heterogeneously among the soil aggregates with a higher intensity downslope than upslope (Figure 8a,b). The reduction in SWR intensity after tree cutting in the *Ch. obtusa* cut tree plot can also support this explanation, as there were no differences in SWR intensity between the *Cr. japonica* intact tree and cut tree plots (Figure 4; Figure 8).

## 5. Conclusions

We conducted a comparative field experiment to interpret the causes of differences in SWR intensity between tree species, cutting, and position on upslope and downslope areas of individual *Ch. obtusa* and *Cr. japonica* trees and stumps. The tree species (*Ch. obtusa*) was identified as the possible main factor increasing SWR intensity. Logging and possible changes in SOM type were identified as two factors affecting SWR reduction. Our results also suggested that SWR intensity increased soil aggregation in the *Ch. obtusa* intact tree plot because of the strong positive correlation between SWR and MWD. Soil aggregate size had negligible effects on SWR intensity; however, the reduction in SWR after breaking up of soil aggregates suggested a possible linkage between SWR and soil aggregate stability. SWR generation (i.e., hydrophobic substances) is an important factor affecting the water repellency of soil, whereas alterations in the SWR intensity are likely as a consequence of physical activities at the soil surface. Nonetheless, SWR is one of the causes of the differences in prevention of water input and low soil hydraulic conductivity between these two species, but the spatial pattern of water movement around a standing tree is likely to be controlled by a combination of other factors.

**Author Contributions:** Data curation, M.F. and K.M.; Formal analysis, M.F. and K.M.; Funding acquisition, A.K.; Investigation, M.F., K.M. and K.O.; Methodology, M.F. and K.M.; Project administration, A.K.; Resources, K.O.; Supervision, K.S.; Writing—original draft, M.F.; Writing—review & editing, M.F. and A.K.

**Funding:** Japan Society for the Promotion of Science: JP26292088, JP18H04152.

**Acknowledgments:** We gratefully acknowledge the staff of Kyushu University Forest for helping and supporting our laboratory and field experiments. This study was partly supported by JSPS KAKENHI Grant numbers JP26292088 and JP18H04152. We thank Catherine Dandie, Alex Boon, and Lucy Muir, from Edanz Group (www.edanzediting.com/ac) for editing a draft of this manuscript.

**Conflicts of Interest:** No potential conflict of interest was reported by the authors.

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
