# Peer review of "Factors Determining Soil Water Repellency in Two Coniferous Plantations on a Hillslope"

_forests, doi:10.3390/f10090730_

Round 1

Reviewer 1 Report

The manuscript presents a very interesting experimental study on soil water repellency, a critical issue for understanding and predicting overland flow and erosion afforested areas. Addressing the effect of clear cutting on soil water repellency and on its spatial variability adds a novel insight on the topic. The species selection is fully justified by their relevance in Japan and the comparison of two coniferous to this respect helps enlarging the set of field data on this tree species class, the most reported in what concerns soil water repellency.

The manuscript is well written and structured, with clearly defined objectives and a well elaborated introduction.

The experiment seems to have potential to provide more relevant results  than those explored by the authors and this is a main drawback of the manuscript. Authors are encouraged to improve data exploration in the revised version of the manuscript.

For this, authors are invited to consider the comments provided long the manuscript.

They regard Materials and Methods, where a better justification of statistical analysis is required. Furthermore, an clarification and justification of soil sample depths is also required, as authors in some parts of the text consider and in some others discard soil layers below 5 cm.

They also regards the Results and Discussion sections. Some indications are made in the comments along the manuscript that are intended to guide authors in a deeper and different exploration of their data.

Author Response

We would like to express our deep gratitude for giving us the opportunity to revise our manuscript. In this revised manuscript lines were changed, we also changed figure numbers. Therefore, all responses are based on new changes. Changes in the manuscript were made with different coloring for reviewers (Red: reviewer #1, Blue: Reviewer #2)

Reviewer #1

The manuscript presents a very interesting experimental study on soil water repellency, a critical issue for understanding and predicting overland flow and erosion afforested areas. Addressing the effect of clear cutting on soil water repellency and on its spatial variability adds a novel insight on the topic. The species selection is fully justified by their relevance in Japan and the comparison of two coniferous to this respect helps enlarging the set of field data on this tree species class, the most reported in what concerns soil water repellency. The manuscript is well written and structured, with clearly defined objectives and a well elaborated introduction. The experiment seems to have potential to provide more relevant results than those explored by the authors and this is a main drawback of the manuscript. Authors are encouraged to improve data exploration in the revised version of the manuscript. For this, authors are invited to consider the comments provided long the manuscript. They regard Materials and Methods, where a better justification of statistical analysis is required. Furthermore, a clarification and justification of soil sample depths is also required, as authors in some parts of the text consider and in some others discard soil layers below 5 cm. They also regard the Results and Discussion sections. Some indications are made in the comments along the manuscript that are intended to guide authors in a deeper and different exploration of their data.

Response: Thank you for your thoughtful comments and positive suggestions. Based on your comments and suggestions, we modified our manuscript as follow:

Q1: The depth column can be discarded from the table and an indication on sampling depth can be provided in table caption.

Response: Thank you very much for your suggestion. Depth column removed from table 2 and added in the caption.

Q2: In the previous subsection it is stated that soil samples were taken at 0-5cm depth. Table 1 presents analysis results for this depth. Yet, in this subsection and in Fig 1, other soil depths are indicated. The following subsections also refer to 3 soil depths. Please clarify and correct accordingly.

Response: Thank you very much for your keen notice. SWR intensity tested in different soil depths to understand its concentration in soil profile. We used this result to explain the differences between two tree species soil and soil surface contamination by probable water repellent materials from Ch. obtusa species. Soil 10 and 30 cm depth removed from Fig 1. Some clarifications added to sentences in Material and Methods part (L101, L102, L122, L123).

Q3: SE stands for standard error? If so, please mention it and also later on, in the Results section.

Response: Thank you very much for your keen notice. SE means standard error in this study. “standard error” mentioned in the first appearance in manuscript (L188) as well as in result part.

Q4: Comments on statistics should be considered relevant by the authors, in order to improve the manuscript. Firstly, authors do not have to refer to ANOVA in one hand and GLM in the other. Simply mention ANOVA and the mean multiple comparison method when applicable (Tukey in this case). Secondly, as in this study only two "treatments" or alternatives are compared, the test is not applicable because the ANOVA identifies the significance of differences between the alternative means. So, in Material and Methods, reference to Tukey should only be mentioned on account of soil depth (3 alternatives compared). Thirdly, and more important, authors performed only one-way ANOVAs. In fact, this is understandable due to the high number of factors involved in the experiment (species, cutting, position, depth). The most appropriate approach would be a 4-way ANOVA, or a 3-way ANOVA when only the upper soil layer is considered in the analysis. This would help identifying the non-significant effects and group data on a second approach, which would an n-1 way ANOVA. Authors should address this topic in Materials and Methods, explaining why they preferred to perform only 1 way ANOVAs.

Response: Thank you very much for your concern about statistical analyses. Generalized linear modeling (GLM) applied to fit the best model (Poisson distribution and a log link) to address which factors (tree species, cutting, position) affected on SWR intensity. After realizing Ch. obtusa is responsible for SWR intensity we tested the one-way analyzing of variance (ANOVA) between upslope and downslope areas in four treatments (see Figure 4b) separately. For SWR in different soil depth, different soil aggregate size and broken soil aggregates, ANOVA applied to the data with 5 (point scale) and 10 (plot scale) replication to detect the difference and when we detect differences post-hoc test (Tukey’s honestly significant differences) performed (L188–L195).

Q5: This is a general comment on terminology applying to the whole text nut in particular to this section. % Ethanol is the expression of results of MED test and, this is taken as an indicator of SWR intensity. Soil water repellency is the key parameter in this study. As so, authors should stick on SWR intensity in the whole text after Materials and Methods. In Materials and Methods, they should explain the connection between SWR intensity, MED test and $ Ethanol. After, these should be avoided. 3 terms to mean the same generate confusion when reading the manuscript.

Response: Thank you very much for you concern about SWR concept. We modified MED test explanation by adding more understandable details with a visualized drawing (Figure 2, L140–L145).

Q6: ANOVA and GLM are not needed after Figure 3. See next comment.

Response: Thank you very much for your comment. ANOVA and GLM removed from result part and graphs as they explained in Material and methods.

Q7: Comments on Fig 3: 1-To which soil depth these results correspond? Please indicate. 2-Commonly, statistical differences between "treatment" means is indicated by letters, starting with a for the highest mean. However, as only 2 "treatments" in each studied factor exist, a simple indication on the category axis could be enough to identify which factor showed significant differences (*) or non-significant differences (ns) between "treatment" means. The actual form overloads the graph area and might reduce legibility. 3- The actual fig title is not correct or it is incomplete. Also, denote the 2 graphs as Fig 3a and Fig 3b as they correspond to 2 very different ways of presenting the same results. Correct Fig title accordingly. 4-Use the same words in the vertical axis and in the Fig title. Then explain in this the meaning of % Ethanol. In the horizontal axis, if the Fig title is adequately stated, there is no need for indicating the statistical test. 5- In Fig title, the part of the text that is not strictly the Fig title should be placed in a Note, furthermore, respectively should be added at the end of the actual Fig title text. 6-Error bars on columns correspond to standard deviation or standard error? Please indicate in Fig title.

Response: Thank you very much for your fruitful comments and suggestions regard to Fig 3 (change to Figure 4 now).

Soil depths added to Figure 4 caption. 2. Based on your comment and Reviewer #2, we decided to used “small letter” and “ns” for showing significant and non-significant differences (we used consistently in all figures). 3. Each graph labeled with letters (a, b, c, d) (we used this method consistently in all figures). 4. Figure 4 caption rephrased. The concept of ethanol % and SWR intensity explained in Materials and Methods clearly (Figure 2 and section 2.3). 5. Unnecessary information removed from Figure 4 and replaced in Figure caption and respectively added as well. 6. Error bars indicate standard error and mentioned in Figure caption.

Q8: See comments on the previous Fig and correct this one and the following accordingly. In this case, it is clear that letters in the columns would better stress differences between "treatments". Better denote Fig 4a - Fig 4d.

Response: Thank you very much for your suggestions. Figure caption corrected according to comments. “small letter” and “ns” used for significant and non-significant differences. Each graph labeled with letter (a, b, c, d).

Q9: The Fig requires a legend to indicate the meaning of black and white colors in columns, instead of explaining this in the Fig caption. Mention to soil depth is required.

Response: Thank you very much for your comments. Legend added into the figure and soil depth mention in caption.

Q10: Authors are not fully exploring the results. For instance, the field measured soil moisture content and its response to rainfall (Fig 6) clearly match Fig 3 left. Please elaborate on these evidences.

Response: Thank you very much for your attention. Fig 3 and 6 changed to Fig 4 and 7, respectively. More detailed and the connection between two figure explained in result part (L234–L237).

Q11: Aggregate mean weighted diameter. Mention to soil depth is required.

Response: Thank you very much for your comments. The correction made in Table 2.

Q12: ANOVA could also have been performed for these soil properties. The ANOVA results could improve the explanations provide by the authors. Note that the correlation analysis (Table 3) was performed for each group. It does not give an insight on the effect of the factors combination represented by each group of data.

Response: Thank you very much for your suggestions. Soil properties results of upslope and downslope and mean of upslope and downslope added into the table 2 and 3 to improve the concept of research and discussion part. ANOVA performed to table 2 data but removed from results to reduce complexity of results for readers. The correlation analysis also performed for upslope and downslope areas to elaborate results.

Q12: See previous comment. These results would be expectable if the sampled areas are homogeneous as regarding soil properties and behavior, which is also assumed on the sampling design. Most certainly, this was not the correlation analysis authors intended to perform in order to proceed towards the paper objectives.

Response: Thank you very much for your notice regards to correlation analysis. The correlation analysis also performed for the data of upslope and downslope areas to bold and explain research objective in result and discussion parts.

Q13: Not significant differences but significant correlation

Response: Thank you very much for your attention.

Q14: Remove last sentence. Information already in Graph legend

Response: Thank you very much for your suggestion. The last sentence removed from Figure 7 caption.

Q15: It is difficult to understand this paragraph. The two last sentences are redundant, considering previous results presented, as they do not regard aggregate size distribution but the effect of species and tree cutting. The first sentence is intended to state that the general decline on SWR with the increase of aggregate size (except for the upper graph left) is not significant? Is this the idea? Please rephrase.

Response: Thank you very much for your consideration. This sentence explained soil different soil aggregate sizes had insignificant impact on SWR intensity weather this soil aggregates located in intact tree plot or cut tree plots. The results of SWR intensity in different soil aggregate sizes showed that SWR intensity are likely related to the type of tree species as the soil aggregates on the downslope of Ch. obtusa individual trees had more SWR intensity than that upslope as well as individual stumps. The results of SWR intensity in soil aggregates in Ch. obtusa plots showed the higher coverage of soil aggregate with water repellent materials than that Cr. japonica.

Q16: Replace last sentence by a graph legend and a rephrased Fig caption (referring to a and b).

Response: thank you very much for your suggestion. The legend added and Fig caption rephrased as a, b, c, and d.

Q17: If statistical exploration of data was differently conducted, authors would reach these same conclusions? Please look upon MWD relation with SWR. Also, separate MWD data on up and downslope subgroups and see if it improves the relationship with SWR.

Response: The relationship between MWD and SWR analyzed by dividing data on upslope and downslope areas and results explained and discussed (L239–L242, L323–L329).

Q18: Do you mean that infiltration would be higher in that case? How this matches with overland flow generation and erosion in clipped areas? Below, the exposure of clipped areas to solar radiation is explored. Why not exploring it also in what regards rainfall amounts reaching the soil surface. Please elaborate more on this.

Response: This means the heterogeneity of infiltration might reduce after removing source (tree) and soil became wet more equally such as Fig 7c. This also explained after loading of water repellent materials into soil nearby stumps stopped, soil expose to outside disturbances. Soil surface might erode within their water repellent materials that cover soil aggregates. Therefore, infiltration started to increase by passing more time as there is no reload of water repellent materials. We did not measure stemflow, throughfall, and gross rain fall in this study. We only used the data of the nearby meteorological station to explained changes in soil moisture after rain event. It is difficult to elaborate the amount of rain reaching soil surface within one single rain event.

Q19: Regardless other effects indicated by the authors in this paragraph, which are valuable explanatory contributions, please consider also the above comments on the correlation analysis presented.

Response: Thank you very much for your suggestion. The correlation analysis also performed on the upslope and downslope areas and relevant discussion added (L323–L329).

Reviewer 2 Report

Please find my specific comments in the attached file.

Author Response

We would like to express our deep gratitude for giving us the opportunity to revise our manuscript. In this revised manuscript lines were changed, we also changed figure numbers. Therefore, all responses are based on new changes. Changes in the manuscript were made with different coloring for reviewers (Red: reviewer #1, Blue: Reviewer #2)

Reviewer #2

We appreciate your positive comments and the opportunity for polishing our manuscript.  Based on your comments and suggestions, we revised our manuscript as follows:

Q1: Please, write the P value

Response: Thank you very much for your attention. P-values added (L18–L20).

Q2: Why do you talk about the hydrophobic substances if you have not analyzed them?

Response: Thank you very much for your keen notice. We considered that hydrophobic substances are responsible for SWR and used this concept to explain and elaborate our results. We replaced hydrophobic substances from the abstract (L21).

Q3: This first part of the introduction does not have a direct relationship with the goal of this work. In this work, there is not any analysis of the hydrophobic substances that may generate soil water repellency. Therefore, you can remove this part.

Response: Thank you very much for your suggestion. We tried to explained and defined SWR with detailed information for those readers who are not familiar with topic in the first paragraph of introduction. We removed this section from introduction and explained only related issue to current research.

Q4: There are more recent works, such as: Jiménez-Morillo et al., 2017. Environmental Research, 159: 394-405. DOI: 10.1016/j.envres.2017.08.022. Benito et al., 2019. J. Hydrol. Hydromech. 67: 129–134. DOI: 10.2478/johh-2018-0038

Response: Thank you very much for your fruitful recommendation. The recommended articles carefully read and used to improve manuscript.

Q5: What is the link between the heat (I suppose that you are thinking in a wildfire) and this work? In this work, there is not any reference to the study of the fire effect in the hydrophobicity.

Response: Thank you very much for your thoughtful question. This sentence removed from manuscript in according to your comment. We mentioned the impact of heat on SWR as it’s an issue in dry areas where wildfire occurs usually. Next, we explain SWR in locations such as UK, Japan and etc. where have humid climate but have dry and wet cycles. Hence, heat is fundamentally a base for enhance SWR either through wildfire or dryness period.

Q6: I think that soil organic matter, soil aggregate size and soil water content are soil properties.

Response: Thank you very much for your notice.

Q7: This is one of the most important sentence of this work.

Response: Thank you very much for your attention.

Q8: Could you mention some of them?

Response: Thank you very much for your recommendation. Examples were added (L62).

Q9: I recommend reading this work, it is very informative: Zavala et al., 2009. Geoderma, 152: 361-374. DOI: 10.1016/j.geoderma.2009.07.011

Response: Thank you very much for your informative recommendation. With all respect to your opinion, Zavala et al. (2009) is related to land use and difficult to connect it to micro-scale study like ours.

Q10: Could you write the coordinate of samples, soil type (Soil Taxonomy) and the climatic conditions? On the other hand, how long did it take since the trees were cut until the samples were collected?

Response: Thank you very much for your suggestions. More site description added in Materials and methods. The current study conducted one year after tree cutting (L86, L88–L93).

Q11: If you only have taken samples in the first 5 cm, how have you analyzed your samples to others depths?

Response: Thank you very much for your keen attention. We evaluated factors determining SWR intensity in 5 cm soil. To understand the concentration of SWR in soil profile we investigate SWR in different soil depths. Fig 1 was modified and clearer information added in Materials and Methods (L101, L102, L122, L123).

Q12: Remove the double space.

Response: Thank you very much for your keen notice. The double space was removed.

Q13: You have to describe shortly the methods that you have used to study the soil properties.

Response: Thank you very much for your recommendation. The methods were used for each soil properties experiments added in Materials and Methods (L105–L108).

Q14: Why did you point in the figure 1 the depth of 30 cm? Have you analyzed the soil at this depth? In the results section you talk that you have analyzed the intensity of SWR at 10 and 30 cm of depth. Nevertheless, you do not say anything of this in material and method section.

Response: Thank you very much for your attention. Figure 1 edited and 10 and 30 cm soil depth were removed. SWR explored in different soil depths to understand its distribution throughout soil profiles (L101, L102, L122, L123).

Q15: This is the only place where you talk of depths in this section, nevertheless, you do not say what were the depths that you sampled.

Response: Thank you very much for your consideration about soil depths. The explanations added to clarify the reason of this experiment (L101, L102, L122, L123).

Q16: This part should be rearranged.

Response: Thank you very much for your comment. The corresponding sentence rearranged with clearer information (L122, L123).

Q17: I am not in agreement with this affirmation. With MED analysis you assess the concentration of MeOH not the penetration/infiltration time.

Response: Thank you very much for your keen notice. The corresponding sentence meant after dropping solution (we used EtOH not MeOH) onto soil surface 3 s were recorded to see whether solution droplet infiltrate within this time or not. This section modified vigorously to be easy for readers. Figure 2 also added to visualized MED test (L131, L132, L140–L145).

Q18: why have you chosen these different concentrations? You have to put the reference.

Response: Thank you very much for your comment. The proper references added (L138).

Q19: With MED you have to recorded the MeOH concentration not the infiltration time.

Response: Thank you very much for your comment. We used different concentration of ethanol in our experiment to understand the potential SWR intensity of samples. To understand this, we have to record a limited time (3 s) for each individual droplet to see whether droplet infiltrate or not. The 3 s selected as ethanol easily evaporate in longer times. The corresponding sentence removed from manuscript as it rose confusion for readers.

Q20: You have not defined SE before.

Response: Thank you very much for you comment. SE defined as standard error (L188).

Q21: Change (40) to [40].

Response: Thank you very much for your keen attention. (40) refers to samples numbers and changed to “(sample number = 40)” (L190).

Q22: I do not understand this sentence. Please check the punctuation symbols.

Response: Thank you very much for your suggestion. The corresponding sentence edited (L199–L201).

Q23: This sentence is very confusing. Could you rewrite it?

Response: Thank you very much for your comment. The corresponding sentences replaced with more readable sentences (L202–L204).

Q24: Please, write a number for each graphic to be able to separate them.

Response: Thank you very much for your helpful suggestion. Numbers added in each graph to ease reading of Figure 4.

Q25: The size of the letters is small.

Response: Thank you very much for your notice. The size of letter enlarged in Figure 4.

Q26: Please rewrite this part, I am not able to understand it.

Response: Figure 4 caption replaced with more readable one.

Q27: Is there significant differences between 10 and 30 cm in Obtusa?

Response: Thank you very much for your thoughtful question. SWR intensity had no significant difference between 10 and 30 cm soil depth. SWR intensity was larger in 5 cm soil depth than that 10 and 30 cm.

Q28: Please, write a number or letter to identified easily each figure.

Response: Thank you very much for your fruitful suggestion. Capital and small letter in each figure denote different graphs and significant differences, respectively.

Q29: What is the value of the depth samples? Please, reedit all figures

Response: Thank you very much for your comment. Figure 5 edited and “0” added to depths that had this value.

Q30: I do not understand why you use this table if you do not say nothing about these values in the text.

Response: Thank you very much for your comment. Table 2 modified into new way of representing data. The appropriate explanations added (L226–L229, L242, L328–L329).

Q31: Table 3

Response: Thank you very much for your notice. Table 2 changed to Table 3

Q32: The differences may be more appreciable if you merge both graphics in one.

Response: Thank you very much for your suggestion. We really respect to your opinion about Figure 8 but we decided to show upslope and downslope data separately. In this way, the concept of our research and last hypothesis enable to deliver to readers more easily.  

Q33: Please, not use the solid line to link the plot points, because this is not a function equation, but they are discrete values. 

Response: thank you very much for your helpful suggestion. Solid lines were removed from Figure 8.

Q34: Could you point out some of them?

Response: Thank you very much for your comment. The relevant factors maintaining SWR intensity are type of tree species, forest operation, slope, soil properties that we evaluated in this study (L272, L273).

Q35: Here you only have studied two tree species.

Response: Thank you very much for your comment. The “(i.e. Ch. obtusa)” added to the sentence to clarify the meaning (L274).

Q36: I do not understand this sentence.

Response: Thank you very much for your comment. The sentence edited to clear the meaning (L278–L280).

Q37: Please, could you say when the sampling area has suffered logging activities?

Response: Thank you very much for your attention. The logging conducted in early 2016 and this study in 2017 (around one year after cutting). The appropriate explanation added in Material and Methods and Discussion parts (L88–L90, L285, L286).

Q38: This affirmation should be supported by specific literature.

Response: Thank you very much for your suggestion. To our knowledge there is no similar study that evaluated the impact of cutting or soil erosion in SWR intensity in soil aggregates. We conduced breaking soil aggregate experiment to support our results in Figure 4 and 6 (L288, L289).

Q39: There are many references.

Response: Thank you very much for your recommendation. The most relevant references used and other eliminated (L291).

Q40: This part is not relevant.

Response: Thank you very much for your suggestion. This part deleted from conclusion part.

Q41: It is very controverted talk about the hydrophobic substances in conclusions if you have not studied them in the work.

Response: Thank you very much for your kind suggestion. The “hydrophobic substances” removed from conclusion and replaced with more appropriate phrases (L349, L350).

Round 2

Reviewer 1 Report

Authors presented a substantial and clarifying response to questions raised during the revision of the original manuscript.

Authors provide a revised version of the manuscript which is recommended for publication in the present form.

Reviewer 2 Report

After check that the changes proposed for the improvement of this paper have been done, I recommend its publication